# Multidimensional Phylogenetic Metrics Identify Class I Aminoacyl-tRNA Synthetase Evolutionary Mosaicity and Inter-Modular Coupling

**DOI:** 10.3390/ijms23031520

**Published:** 2022-01-28

**Authors:** Charles W. Carter, Alex Popinga, Remco Bouckaert, Peter R. Wills

**Affiliations:** 1Department of Biochemistry and Biophysics, University of North Carolina at Chapel Hill, Chapel Hill, NC 27599-7260, USA; 2Centre for Computational Evolution, University of Auckland, PB 92019, Auckland 1142, New Zealand; alexpopinga@gmail.com (A.P.); remco@cs.auckland.ac.nz (R.B.); 3Department of Physics and Te Ao Marama Centre for Fundamental Inquiry, University of Auckland, PB 92019, Auckland 1142, New Zealand; p.wills@auckland.ac.nz

**Keywords:** BEAST2, DensiTree, protein mosaic structure, RNA World hypothesis

## Abstract

The role of aminoacyl-tRNA synthetases (aaRS) in the emergence and evolution of genetic coding poses challenging questions concerning their provenance. We seek evidence about their ancestry from curated structure-based multiple sequence alignments of a structurally invariant “scaffold” shared by all 10 canonical Class I aaRS. Three uncorrelated phylogenetic metrics—mutation frequency, its uniformity, and row-by-row cladistic congruence—imply that the Class I scaffold is a mosaic assembled from successive genetic sources. Metrics for different modules vary in accordance with their presumed functionality. Sequences derived from the ATP– and amino acid– binding sites exhibit specific two-way coupling to those derived from Connecting Peptide 1, a third module whose metrics suggest later acquisition. The data help validate: (i) experimental fragmentations of the canonical Class I structure into three partitions that retain catalytic activities in proportion to their length; and (ii) evidence that the ancestral Class I aaRS gene also encoded a Class II ancestor in frame on the opposite strand. A 46-residue Class I “protozyme” roots the Class I tree prior to the adaptive radiation of the Rossmann dinucleotide binding fold that refined substrate discrimination. Such rooting implies near simultaneous emergence of genetic coding and the origin of the proteome, resolving a conundrum posed by previous inferences that Class I aaRS evolved after the genetic code had been implemented in an RNA world. Further, pinpointing discontinuous enhancements of aaRS fidelity establishes a timeline for the growth of coding from a binary amino acid alphabet.

## 1. Introduction

The emergence of the aminoacyl-tRNA synthetases, aaRS, is a quintessential chicken and egg puzzle whose solution would demystify the origins of coded protein synthesis. How did aaRS enzymes gain the reflexive property of collectively being able to use relationships in the universal genetic code to convert the sequences of base triplets in their own genes into functional amino acid sequences that make the code work? The detailed trajectory by which genes for the two essential superfamilies appeared, acquired catalytic proficiencies and radiated to refine their dual amino acid and cognate tRNA specificities is thus a crucial chapter in the book of life. 

Both aaRS Classes [1,2,3] have separate catalytic and anticodon-binding domains. Only the catalytic domains within each superfamily share the same architectures [4]. Anticodon-binding domains are idiosyncratic and, by consensus probably have a distinct evolutionary origin [5]. We develop a quantitative framework of high-resolution phylogenetic metrics here to show that even the shared architectures of the large, variable catalytic domains are probably mosaics assembled from smaller peptides. 

Phylogenetic clades for each amino acid in the Class I aaRS superfamily tree [6,7,8,9,10,11,12] are monophyletic and divide into three subclasses [4,13]. Subclass IA includes IleRS, ValRS, LeuRS, and MetRS; subclass IB GluRS and GlnRS; and Subclass IC TyrRS and TrpRS. CysRS and ArgRS, assigned originally to Subclass IA [13], are difficult to assign, with one or the other appearing instead with Subclass IB (for example, see [9]). 

Genetic deconstruction and experimental biochemistry have suggested significant mosaicity within the Class I aaRS catalytic domains (Figure 1). Two segments nested within these domains—protozymes (from προτο = first [14]) and urzymes (from Ur = original, authentic [15,16,17])—represent successive intermediate evolutionary states of increasing length and dating from well before the Last Universal Common Ancestor [18,19]. 

Urzymes [25] are catalytically active cores whose ~130 residues form a nearly intact active site within the ~250-residue catalytic domains of Class I and II aaRS. Class I urzymes are excerpted from the full length enzymes and rehabilitated by protein design [26] to reconfigure the disrupted solvent accessible surface exposed upon deleting a major part of the catalytic domain and to seal the gap between two disjoint parts of the active site left by removing “connecting peptide 1” (CP1; [20,27]) (Figure 1B). Urzymes approximate the full extent of bidirectional genetic encoding opposite Class II urzymes [16]. Biochemical experiments show that urzymes retain ~60 % of full-length aaRS catalytic proficiency—estimated as the transition-state stabilization free energy—for both amino acid and tRNA substrates [16,26,28]. Class I and II urzymes also differentiate between the corresponding two sets of substrates; Class I urzymes activate Class I, in preference to Class II amino acids by ~1.0 kcal/mole, and conversely [23,29,30]. Thus, they retain the functionality expected of evolutionary ancestors.

Protozymes are 46-residue subsets from both Class I and II Urzymes that retain nearly half the full catalytic proficiency in the critical amino-acid activation reaction, whose uncatalyzed rate is rate-limiting for protein synthesis [14]. The Class I protozyme coincides with the N-terminal crossover connection of the Rossmann fold, which forms the ATP binding site and contains a distinctive 3D packing motif that recurs in ~25% of the proteome [31] and imposes distinct, functionally relevant conformational states tightly coupled to catalysis in TrpRS [24,32,33,34,35,36,37]. As discussed below, presence of the ATP binding site in the protozyme justifies expectations that its sequences may be less variable than those involved in amino acid binding, as well as increasing the likelihood that it may represent a more ancient peptide module than the full urzyme sequence. 

The Class I protozyme is N-terminal in Class I, but C-terminal in Class II urzymes, in keeping with the Rodin-Ohno hypothesis of ancestral bidirectional coding [22]. Fully active Class I and II protozymes have been expressed from a single gene designed to encode their structures on opposite strands. Experimental Michaelis-Menten parameters of all four protozymes—Class I and II; native and bidirectional—are, remarkably, the same within experimental error [14]. Tamura’s laboratory [38] have replicated those results. Experimental [14,17,39] and bioinformatic evidence [40] therefore support the hypothesis of Rodin and Ohno [22,41] that the two aaRS Classes descended from opposite strands of a single bidirectional gene. 

Structural conservation, high catalytic proficiency in both essential reactions, and specificity for amino acids from the appropriate class all reinforce their role as experimental for different stages of aaRS molecular ancestry.

CP1 intersects the Class I aaRS Rossmann fold immediately after the protozyme between structurally homologous residues that are ~4.5 Å apart. CP1 can thus be replaced by a peptide bond in all Class I aaRS, without structural disruption to produce the Class I urzyme [15,17] as detailed in Figure 1. CP1 insertions include the editing domains of the aaRS for aliphatic amino acids, and thus largely account for the variable size of Class I catalytic domains. Independent 3D superposition of aaRS crystal structures [6,17,42] revealed considerable structural conservation within the catalytic domains of all ten members of each Class—see Figure 1 of [42]. Surprisingly, structural homology across the Class I superfamily extends beyond the urzyme, into CP1. We call the secondary structures within these conserved cores “scaffolds”. 

Evidence for descent of Class I and II aaRS from a bidirectional gene implies, *ipso facto*, that sequences inconsistent with bidirectional coding, like CP1, represent accretion of new genetic material. The CP1 insertions are incompatible with, and their introduction would necessarily have ended, bidirectional genetic coding of ancestral Class I and II aaRS (Figure 1B). Class I urzyme boundaries delimited decisively by potential bidirectional coding of Class II urzymes constitute only about 80% of the Class I scaffold; the remainder consists of 10-residue α-helical segments near the beginning and end of the CP1 insertion (Figure 1). Thus, if the Rodin-Ohno hypothesis is correct, then the CP1 insert must derive from a distinct, more recent genetic source. 

To address the apparent contradiction between the Rodin-Ohno hypothesis and extended conservation into the CP1 segment, we assembled phylogenetic metrics that, together, reinforce the conclusion that modular components within the structurally invariant segments of the ancestral aaRS have different genetic histories: (i)We threaded sequences into closely-related crystal structures to assemble multiple sequence alignments (MSAs) based on three-dimensional structure superposition, to avoid using amino acid substitution matrices to define equivalent positions.(ii)We compiled multi-dimensional phylogenetic metrics from the ensemble of phylogenetic trees obtained by Markov Chain Monte Carlo (MCMC) simulations.(iii)We increased the effective analytical resolution by applying the metrics to partitions of the MSA that have been extensively characterized experimentally, providing novel insight into the functional modularity of the Class I aminoacyl-tRNA synthetase superfamily.(iv)We identified two-way interactions between mutation rates in different MSA partitions. Both involve the amino acid binding site and one is central to the amino acid specificity enhancement enabled by the CP1 insertion.

These results support important modifications of conventional evolutionary scenarios for major parts of the proteome containing Rossmann dinucleotide fold domains and strengthen the proposal that the genetic code development is coupled intimately to the structural evolution of aaRS.

## 2. Results

Our focus will be the phylogenetic metrics in the shaded columns in Table 1, derived from trees constructed for MSAs in different rows—CP1; the urzyme; and its three distinct modules, Urz_A (the protozyme), Urz_B (the amino acid binding site), and Urz_C (the PP_i_ binding site). Only three of the five columns are linearly independent. The two clade support columns (${\langle S\rangle }_{WAG}$ and ${\langle S\rangle }_{LG}$) were derived using the WAG and LG substitution matrices, without allowing changes in the multiple sequence alignments during the MCMC searches, and so are nearly identical. We show below that <Q> is linearly dependent on log(Shape). We consider two novel, complementary implications of these metrics.

The modular MSAs are widely separated in the vector space spanned by the three linearly independent metrics (Figure 2). The CP1 MSA lies on the floor near the front of the 3D plot, whereas those derived from the urzyme cluster in the upper left rear corner, where they form a small triangle (red lines).

Multiple regression analyses demonstrate that the separations between the MSAs in different rows of Table 1 can be rationalized by functional dependence of each metric on the independent variables in the unshaded columns of Table 1. Moreover, these independent variables represent previously recognized structural and biochemical properties of the modular components motivating the partition of MSAs.

We note that these conclusions required only publicly available software (BEAST2, Densitree, and JMP or other standard statistical software), the MSA data are provided in the supplement, and we do not describe any new software platforms. (i) and (ii) together furnish unprecedented insight into the genetic modularity of the Class I aaRS. 

In summary, these metrics suggest that CP1 is derived from a more recent and less cladistically coherent genetic entity, as previously posited [17]. Similarly, they suggest that the protozyme may derive from an older genetic entity than other segments of Class 1 urzymes.

### 2.1. CP1 Has Lower and More Uniform Apparent Site-to-Site Mutation Rates between aaRS for Different Amino Acids

The BEAST 2 MCMC algorithm tracks the extent of site-to-site evolutionary of sequence variation in using the Tree Height and Shape metrics described in Methods. The Tree Height metric is summarized in (Figure 3A,B). Sequences responsible for the elevated urzyme Tree Height are identified by regression against the independent parameters of the design matrix (Table 1) in Figure 3C, using the regression model: Tree Height =2.17−1.4∗Segment B+3.2∗Urzyme−2.7∗protozyme+1.7∗Segment B∗protozyme. All β coefficients have *p*-values < 0.005. Foremost among the positive contributors is the urzyme. However, the interaction between the protozyme (segment A) and segment B is also significant. CP1 sequences exhibit substantially smaller Tree Height and variance (i.e., higher Shape), consistent with a more recent genetic entity. 

The negative logarithm of the Shape parameter is highly correlated with the conservation quality, $\langle Q\rangle$ defined by Clustal [43,44] (Figure 4A), lending intuitive insight into the physical meaning of the latter. The designation “Conservation Quality” is apparently misleading in suggesting that the significantly smaller $\langle Q\rangle \;$value for the CP1 MSA (Figure 4B) implies that it is less well conserved, in apparent conflict with its reduced Tree Height. In fact, the colinearity of –log(Shape) and $\langle Q\rangle$ led us to pursue the equivalence between the two metrics. Figure 4C,D show that the two metrics depend in nearly identical fashion on predictor columns from the design matrix in Table 1. The $\langle Q\rangle$ metric does not measure mutational variation per site itself, but rather the logarithm of its variance.

Moreover, the extended similarity between Figure 4C,D suggest that column-by-column (i.e., site-by-site) metrics (Tree Height and Shape) provide high resolution evidence on the evolution of modularity. 

Regression of Shape on the independent parameters of the design matrix in Table 1 (Figure 5) resulted in the unique model: Shape =1.37+0.4∗Segment B−1.08∗Segment C+1.56∗CP1−0.57∗Segment B∗CP1. All β coefficients are highly significant (*p* < 1 × 10^−5^).

### 2.2. Urzyme-Based Clades Are More Congruent and Monophyletic Than Those for CP1

Urzyme and CP1 partitions of the MSAs produce substantially different trees (Figure 6A). In particular, although all ten Class I aaRS clades are monophyletic in the trees for the urzyme MSAs, three clades in the CP1 trees—MetRS, ValRS, and TyrRS—are polyphyletic. Moreover, the urzyme clades are constrained by tight envelopes, whereas the CP1 envelopes are poorly defined. 

Support, *S_i_*, the fraction of all trees for which each aaRS clade, *i*, is monophyletic, was averaged over all Class I aaRS types to give the mean support, S=∑Si10.

That operation was repeated for trees built for the full scaffold MSA (Full), the urzyme, CP1, protozyme (segment A in Figure 1), and seven subset MSAs each consisting of twenty amino acids in blocks of five residues as described in Methods. For the eleven rows of the design matrix (Table 1), we computed <S> for populations of trees constructed using the conventional WAG [45] substitution matrix and repeated using the more recent LG [46] substitution matrix used in [12]. 

Contributors to the variance of $\langle S\rangle$ were assessed using stepwise multiple regression, which resulted in the unique model: $\langle S\rangle =0.67+0.04*protozyme-0.33*CP1+0.1*protozyme*CP1$ summarized in Figure 6. All three coefficients are significant, with *p*-values < 0.005. As suggested by its position in the regression curve in Figure 6, trees built from CP1 residues have significantly less support than those from the urzyme or any of its 20-residue subsets. The A segment coincides with the protozyme MSA; its dominant impact on the variance of $\langle S\rangle$, contributing positively both via its intrinsic effect and by its two-way interaction with CP1, is consistent with its being close to the root of the Class I aaRS superfamily tree.

Thus, although the 3D structures from the urzyme and CP1 partitions have comparable structural homology, they have markedly different phylogenetic signatures. 

### 2.3. Tree Height, Shape, and Support Reveal Significant, High-Resolution Genetic Mosaicity

Our quantitative evidence probes far deeper into evolutionary time than previous phylogenetic analyses of Class I aaRS [7,11,18,19,47]. That depth both calls for caution and is a source of great interest. Constructing aaRS trees is fundamentally ambiguous because at each node in any conceivable tree venturing beyond nodes at which aaRS for related amino acids merge, the coding alphabet and dimension of the operational substitution matrix necessarily both change by integer steps as Class I and II trees branch into multiple families. Trees for the two superfamilies are thus necessarily interdependent, so that the dimensions of all possible substitution matrices start from 2 and end at 20. Thus, it is uncertain what should be inferred from phylogenetic metrics for a single superfamily. 

These difficulties are substantially offset by the consistency of site-by-site and row-by-row metrics with the construction and experimental characterization of aaRS protozymes [14] and urzymes [23,24,25,28,29,33,39] (Figure 1). Such deep evolutionary intermediates are, at present, manifestly unique to the aaRS. The extensive experimental and structural context of that consistency strengthens our conclusions even without comparable analysis of Class II aaRS, now in progress, especially in light of the following observations. 

#### 2.3.1. The Three Metrics Are Uncorrelated

The CP1 MSA is a substantial outlier for all three metrics, suggesting that the metrics may be correlated. Removing the CP1 entries from the design matrix eliminates any correlation between Tree Height, Shape, or $\langle S\rangle$, Table 2. The three types of metrics are therefore essentially uncorrelated and provide independent insights.

#### 2.3.2. Differences between Urzyme and CP1 Sequences Are Statistically Meaningful

If the Class I aaRS sequence partitions compared in Table 1 were all drawn from continuously replicated ancient genetic sources, subject to comparable selection history since their emergence, the null hypotheses would produce similarly conserved sequences and comparably congruent clades for the urzyme and CP1 partitions. The log-worth values (i.e., –log(P)) for CP1’s higher mutational frequency (Tree Height; 2.7), its variance (Shape; 3.3) and lack of congruence for phylogenetic trees, ($\langle S\rangle$; 3.4), imply with high statistical significance that all metrics for CP1 arise from a different genetic population than the urzyme sequences, corroborating that—based on bidirectional coding ancestry—the CP1 sequences represent genetic information acquired more recently by the urzymes. 

#### 2.3.3. The Lengths of Different Segments Drawn from the MSA Have No Detectable Impact on Any Phylogenetic Metric

One might suppose that degraded congruence of the CP1 trees results from the fact its MSA has only ~0.25 as many amino acids as the urzyme MSA. However, including the NUMB parameter in Table 1 fails to reduce the variance of regression models for any score (Figure 2, Figure 3, Figure 4 and Figure 5). The insignificant impact of NUMB and the clustering of the 20-residue $\langle S\rangle$ values with the intact urzyme MSA (Figure 5 and Figure 6) confute that expectation.

#### 2.3.4. Threading Does Not Force Any Particular Comparison between Different aaRS Types

Threading increased the reliability of structure-based alignments within any aaRS type by adding sequences. Scaffold positions were rigorously defined as structural homologs from the close proximity of their alpha carbon coordinates in multiple PDB structure alignments (i) among aaRS for any single amino acid, drawn from very diverse bacterial species, that only then (ii) produced a grand scaffold MSA across all Class I aaRS types. Additions produced by threading thus have only second-order effects on structural superpositions of aaRS for different amino acids, adding precision to our analysis without overtly influencing choices on which our conclusions depend.

#### 2.3.5. Distinguishing Features of CP1 Sequences Are Evident without Considering Indels

Figure 2, Figure 3, Figure 4, Figure 5 and Figure 6 together illustrate a comprehensive, near-optimally quantitative three-dimensional comparison of structural partitions in the Full Class I aaRS scaffold MSA. The crux of what the data suggest is that CP1 is more recent than the urzyme, its evolutionary divergence and variance—evidenced by its Tree Height and Shape—is much reduced, and its phylogenetic consistency—evidenced by $\langle S\rangle$—is also much reduced, relative to the corresponding metrics for urzyme segments. This counterintuitive conclusion is evident, from sequences with strict 1-1 correspondences between 3D crystal structures, excluding the large and variable-length indels that dominate CP1 insertions in most aaRS types. We consider this key observation in greater detail in the Discussion.

#### 2.3.6. Neither Convergent Evolution nor Horizontal Gene Transfer Is a Likely Explanation for the Urzyme/CP1 Distinction

The congruent clade structure of urzyme-derived sequences from the scaffold separates into consensus groupings characterizing Class I aaRS as a coherent superfamily, the explanation of which does not require reference to mechanisms beyond mutation and selection from a single common ancestor. The aberrant behavior of sequence variations in the CP1 insertion (Figure 3, Figure 4, Figure 5 and Figure 6) might suggest appeal to such processes. Treatments of Class I aaRS evolution based on full MSAs [10,48,49] show evidence of horizontal gene transfer (HGT), genetic transpositions and large scale insertion/deletion events, of which CP1 is the foremost example. 

Wherever full-length bacterial Class I enzymes with a particular amino acid substrate specificity are represented by more than one canonical structure, as has been described for IleRS and MetRS (see Figure 3 in [48]), that bipartite distribution in genome space is adequately explained in terms of early HGT into bacteria from an archaean/eukaryotic ancestor, but not in terms of convergent evolution. CP1 sequence disparities at homologous sequence positions in TyrRS, MetRS, and ValRS behave in the opposite manner: the higher variance of their $\langle Q\rangle$ values producing lower $\langle S\rangle$ values by allowing their evolutionary paths to wander widely, often crossing in sequence space, instead of forming multiple well-defined clades reasonably distant from one another in sequence space that could arise as a result of HGT. 

### 2.4. Phylogenetic Metrics Have Functional Interpretations

The fact that Class I aaRS amino acid binding sites are bounded by segment B and the protozyme (Figure 1) furnishes a glimse into the functional significance of the MSA distribution in Figure 2. The negative β coefficients of these two predictors from the regression in Figure 3 show that these two segments reduce the Tree Height, relative to the positive urzyme contribution, hence increasing the estimated column-by-column mutational frequency within those segments. Regressions in Figure 4, on the other hand, document the opposing effect of the B subset, relative to the urzyme and C subset, on the –log(Shape) and $\langle Q\rangle$ variance predictors. The opposite signs of β coefficients signify that the presence of sequences in segment B sharpen, whereas those in the urzyme and especially segment C broaden, the distribution of mutational frequencies given by the inverse of the Tree Height in Figure 3. Thus, sequences within the amino acid binding site have the highest apparent mutational frequencies with, simultaneously, the tightest distributions. This conclusion, analogous to that illustrated in the histograms in Figure 3A, points to functional relevance at even higher resolution.

Regression models of the three independent metrics derived from BEAST2 tree constructions all depend heavily on significant two-way interactions between segments of the different MSAs. The Class I aaRS modules are experimentally sufficiently well-characterized to sustain functional interpretations of the three two-way interaction terms for site-to-site (Figure 3 and Figure 5) and row-by-row Support (Figure 6) metrics. These interpretations, in turn, shed light on how evolutionary changes enhanced genetic coding.

#### 2.4.1. CP1 Forms a Structural Annulus Constraining the Urzyme’s Two Halves

The simplest of the 10 CP1 insertions (in TrpRS) is only 74 residues long. Its structure wraps around the protozyme and specificity-determining helix on one side of the active-site opening (Figure 7). Molecular dynamic simulations of the TrpRS urzyme [26] show that in these two segments, which together form the Class I aaRS amino acid binding sites, exhibit extensive relative motion. Further, coupled relative motion of CP1 and the anticodon-binding domain reduces the distance between the two parts of the amino acid binding site in the TrpRS catalytic conformational transition [33]. Steady-state kinetic measurements of amino acid specificity [33] confirm that this relative domain motion enhances the relative specificity for Tryptophan vs Tyrosine. This structural feature provides a functional interpretation for how the three two-way interactions (Figure 3, Figure 5 and Figure 6) contribute to the phylogenetic metrics.

#### 2.4.2. Tree Height Dependence on Segment B Changes Sign, Depending on Whether the Protozyme Is Present

As noted in regard to Figure 3A,B, the positive β coefficient for urzyme sequences increases the Tree Height metric, reducing the estimated mutation rate. Protozyme sequences, and those in Segment B have the opposite effect, increasing the estimated mutation rate. A reflection of this phenomenon is that the protozyme and urzyme appear in much the same place on the regression plots in Figure 5 and Figure 6, yet are well separated in Figure 3, where the protozyme is midway between the urzyme and CP1. Further, sequences in Set 1 from within Segment B have the highest Tree Height, i.e., are the most highly conserved. The increased apparent mutation rates of protozyme and Segment B sequences are offset by the coupling between them: its β coefficient, +1.66, is intermediate between their two negative coefficients. 

The coupling in the Tree Height regression model is between the protozyme—locus of ATP binding—and segment B—locus of amino acid binding (Figure 1A and Figure 7B,C). Its β coefficient arises from the contravariant effects of the two different binding sites. The protozyme’s ATP binding site is common to all Class I aaRS. Segment B is part of the amino acid binding site, and thus would be expected to exhibit the most significant evolutionary sequence variation between families specific for different amino acids.

#### 2.4.3. Shape Dependence on Segment B Changes Sign, Depending on Whether CP1 Is Present

The interaction between CP1 and the amino acid binding site of the urzyme constrains the amino acid binding site dynamically. Structural relationships between CP1 and the amino acid binding site in TrpRS are highlighted in Figure 7B. The experimental demonstration of energetic coupling between CP1 and the amino acid binding site validates the sign and strength of the B*CP1 contribution to Shape, much as a pre-formed space in a partially assembled puzzle validates the outline of the missing piece. 

#### 2.4.4. Support Dependence on CP1 Changes Sign, Depending on Whether the Protozyme Is Present

CP1 has the most significant impact on the regression model for the Support metric, $\langle S\rangle$. Its β coefficient is −0.33, more than three times that of the next most important predictor. This effect can be seen in the regression plot in Figure 6, in which the MSA for CP1 is more widely separated from the other MSAs than in the models for any other metric. Residues within the protozyme contribute more decisively to this metric than do residues located elsewhere in the urzyme. Moreover, the presence of the protozyme sequences in the Full MSA is sufficient to change the impact of CP1 from negative to positive, giving the β-coefficient of +0.1 for the protozyme*CP1 interaction term. Protozyme sequences enhance the cladistic coherence of CP1. That two-way coupling, the widespread occurrence of the protozyme packing motif [31], and its likely role in activating ATP [14], reinforce the conclusion that the protozyme was the original root of the entire superfamily and is older than the urzyme itself, as previously proposed [50].

## 3. Discussion

The progressive biochemical functionality of aaRS protozymes and urzymes, the fact that their structures are universally conserved within both Classes, and the evidence that they descended from one bidirectional ancestral gene all imply that they are legitimate experimental models for ancestral evolutionary aaRS forms. They represent assignment catalysts that participated throughout the emergence of genetic coding and well before LUCA [25]. Moreover, to date, no such sequential intermediates have been characterized for other superfamilies. Thus, more appeared to be known about the modular evolution of the two aaRS superfamilies in advance of this work than for any other ancient superfamily, making the Class I aaRS an appropriate subject our study. 

The biological import of our results is to furnish new phylogenetic support for an evolutionary trajectory assembling different polypeptide sequences with successive capabilities necessary for the emergence and refinement of genetic coding (Figure 8): Mobilization of ATP as an energy source for amino acid activation (protozyme)Simultaneous recognition of amino acid and tRNA substrates and a rudimentary binary code (second half of the urzyme)Insertion of an ancestral CP1, perhaps from an RNA transposable element, to produce a rudimentary catalytic domain and terminate bidirectional coding (CP1)Assimilation of idiosyncratic anticodon binding domains (ABD)Expansion of the coding alphabet via mutational generation of allosteric coupling between CP1 and ABD modules and multistage error correction (editing domains).

**Figure 8 ijms-23-01520-f008:**
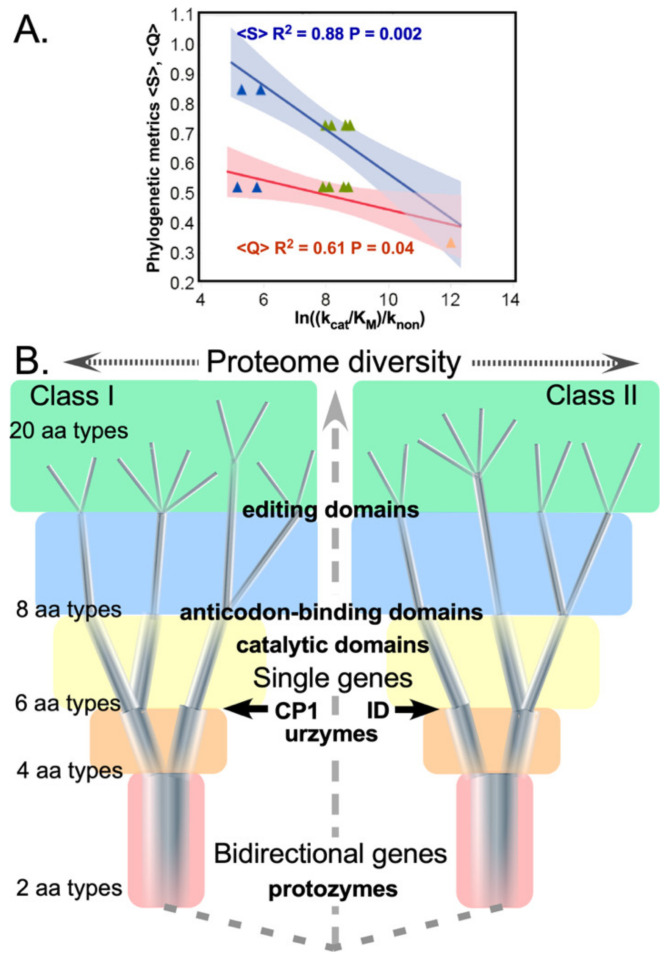
Assignment catalysis and code evolution. (**A**) Correlations between phylogenetic metrics, $\langle Q\rangle$, $\langle S\rangle$, and experimentally determined rate accelerations. Parameters for regression against experimental catalytic proficiencies for corresponding putative evolutionary intermediate constructs (P = 0.04, 0.003, respectively) are both significant. Blue, green, and amber triangles represent protozymes, urzymes, and catalytic domains. (**B**) Timeline for growth of the genetic coding alphabet from a two-letter code. Introducing new aaRS into the context of the ancestral bidirectional gene (red background and dashed connecting lines) simultaneously enhanced specificity and created fundamental changes in selection pressure. Different colored backgrounds signify altered selection pressures that apply to all extant aaRS at a given stage of the coding alphabet, as well as the scale of the extant proteome, possible with successive alphabet sizes. Increasing cardinality of the alphabet induces (i) sequence space inflation, as a greater number of distinguishable sequences can be specified; and (ii) restriction in the diameters of the quasispecies, as they approach fully coded sequences with the final 20-letter alphabet. Boldface landmarks (CP1 and ID) denote qualitative changes in the aaRS architecture shown experimentally to enrich specificity, as discussed in the text. Note that this timeline represents evolutionary events before the LUCA.

### 3.1. Primordial aaRS Quasispecies Covered Progressively Smaller Regions of Sequence Space, Closer to the “Root”

Early aaRS evolution cannot have been an ordinary mutation/selection process. All solutions to the problem normally framed as a "chicken and egg problem” [51] imply historical context and a continuity principle must be defined by the genotype-phenotype mapping [52]: at any stage during the emergence of coding some prior system must have been interpreted by extant aaRS protogene translation products. Consistent with this expectation, the phylogenetic metrics $\langle Q\rangle$ and $\langle S\rangle$ correlate with rate enhancements measured for successive experimental models of ancestral aaRS (Figure 8A). In turn, those proficiencies themselves correlate quantitatively with the concomitant additions of mass in the Class I and II aaRS evolutionary intermediates they entail (see Figure 6 in [25]).

A key aspect not yet explored for the prior systems approximated by those evolutionary intermediates is that as the alphabet size, diversity, and consequent fidelity increased, they would, *ipso facto*, have created more narrowly targeted selection pressures, strongly coupling mutation and selection in early stages of genetic coding (Figure 8B). The subsequent coevolution of CP1 and urzyme sequences would necessarily have preserved urzyme functionality, while the new CP1 could adapt flexibly to its developing role of enhancing specificity as described in the next section.

Different background colors in Figure 8B denote how branching of the tree to allow introduction of the *n*th amino acid into the alphabet enforces a highly cooperative re-optimizing of the *n* − 1 aaRS types already present. As each new, refined amino acid type emerged, all extant gene sequences adapted to opportunities introduced by progressively finer discrimination between amino acid side chain physical chemistry. In turn, adaptation to a more diverse alphabet sharpens the new aaRS fidelities (i.e., the contraction of branch thicknesses in Figure 7B), implicating a bootstrapping feedback and enhancing the cooperativity of the transition to higher-dimension coding alphabets [50]. 

That cooperativity creates a Lamarckian-like correlation between selection pressure and its outcome—the result of any mutation being nearly synonymous with the selection pressure it faced, especially as the code differentiated. Survival would have depended on the relationship between the shapes of fitness landscapes and error rates of catalysis by the extant quasispecies [53]. The earliest alphabets, including at least the first two amino acid types, were also tightly constrained by bidirectional coding (rose-shaded background, Figure 8B). 

We have argued [50,54] that a single functional island in sequence space (i.e., quasispecies) would invariably have been a strong attractor, irrespective of detailed features of the fitness landscape that stabilized it, because all mutations that moved the system slightly off its optimum phenotype would be subject to strongly restorative selection pressures. However, if ancestral protozymes from a bidirectional gene had broad, relatively flat (and necessarily co-dependent) fitness landscapes, matched to correspondingly high error rates, that could have favored bifurcated quasispecies that enhanced genetic coding by recruiting new amino acid types and simultaneously increasing the precision with which child specificities could be encoded. 

### 3.2. Evolutionary Refinements of Protein Catalysis and Specificity Were Predicated on Expanding the Genetic Code

The Tree Height, Shape, and Support metrics identify differences in the genetic origins of successively acquired contributors to aaRS function: protozyme=>urzyme=>catalytic domain with CP1. Both phylogenetic and functional properties of these intermediates thus appear to probe far deeper into evolutionary time than previous phylogenetic analyses of Class I aaRS [7,11,18,19,47]. 

The precision with which protein active sites distinguish substrates from one another, and transition states from substrates, was the result of the evolutionary process we begin here to characterize. Expansion of genetic coding itself depended critically on developing a system for the placement of a precisely defined amino acid sidechain at a particular point on an aaRS peptide backbone. Phylogenetic analysis of segmental Class I aaRS MSAs represents a uniquely promising opportunity to test the hypothesis that contemporary enzymes are mosaic structures rooted in simpler catalytic polypeptides and assembled from detectably different genetic ancestors. 

### 3.3. Phylogenetic Metrics Identify Meaningful Fine Structure and Covariation within Multiple Sequence Alignments

Section 2.2, Section 2.3 and Section 2.4 exploit quantitative metrics compiled during the Markov Chain Monte Carlo exploration of the phylogenetic landscape to expose differences in how distinct segments of the overall MSA behave. The statistical coherence of these phylogenetic metrics alone justifies their novel application here. Their functional significance emerges only in the context of partitioning the overall MSA according to structural and functional criteria established within other disciplines and in the presence of suitable controls for the effect of sequence length (subsets 0–6 from the urzyme MSA; Figure 1C, Table 1). 

In turn, the phylogenetic analysis provides novel validation of decisions that guided experimental work on partitioning the MSA. Identification of how the Tree Height (Figure 3B), Shape (Figure 5), and Support (Figure 6B) depend on two-way intermodular interactions validates experimental work by demonstrating how inter-modular coupling contributes to catalysis and specificity. Further, because these interactions arise from the coherence across the Class I superfamily, they imply that similar interactions occur between ATP and amino acid binding sites and between amino acid binding sites and CP1 in all or most Class I aaRS.

### 3.4. High-Resolution Structural Modularity Implies Discontinuities in the Evolution of Genetic Coding

If CP1 insertions were indeed assimilated from one or more similar genetic sources after Class I aaRS urzymes had evolved significantly from ancestral protozymes, it could have at least three noteworthy biological implications: (i)Class I protozymes, whose catalytic activity mobilizes ATP for biosynthesis—an activity found in many proteins—may root the substantial portion of the proteome built from β-α-β crossover connections. That portion would include the entire Rossmannoid protein superfamily [31] and potentially β-barrel proteins [55,56], which are central to intermediary metabolism and nucleotide biosynthesis [57]. (ii)AARS protozyme and urzyme populations would have functioned first as quasispecies in translation, limiting the sophistication of the early proteome.(iii)CP1 assimilations would have transformed selection pressures for subsequent aaRS evolution by facilitating enhanced fidelity.

### 3.5. Insertion of CP1 Likely Enabled Saltatory Improvements in Fidelity

Structural and biochemical data suggest that the CP1 insertions created stepwise enhancements in the evolution of genetic coding by enabling conformationally-driven mechanisms to increase specificity. The shortest CP1 insertions have ~75 residues in TrpRS and TyrRS that recur essentially intact in the longer CP1 insertions of the remaining eight Class I aaRS [17,42]—enabling our identification of the Class I scaffold. It seems likely that the initial insertion needed to be that long. CP1 must wrap around the urzyme (Figure 7A) to constrain relative movements of the protozyme and specificity helix that form the amino acid binding site [26]. For that reason, an earlier hypothesis as to its origin referenced near-simultaneous insertion of a mobile genetic element into all extant Class I urzymes [17], in which case its root sequence would have been more recent than that of the urzymes, yet earlier than the remaining sequences in contemporary full-length aaRS enzymes. 

Amino acid specificities [23,29] suggest that although capable of 10^9^-fold rate accelerations, Class I and II urzymes select an amino acid from the correct Class only 80% of the time. However, Wills & Carter [30] note that within-Class aaRS urzyme specificity is consistent with each Class distinguishing two kinds of amino acids, to operate a four-letter coding alphabet. These modest fidelities suggest a fundamental limit to the precision of which bidirectional coding was ultimately capable. 

The most evident contribution of CP1 to fidelity is that the editing domains present in the larger subclass IA aaRS for aliphatic amino acids Ile, Val, and Leu are elaborations of the CP1 motif present in the simplest subclass IC aaRS for tyrosine and tryptophan. It seems likely, however, that CP1 functioned even earlier to enhance fidelity by dynamically constraining the volume and configuration of the amino acid binding pocket. Several groups found that comprehensive mutation of side chains in the immediate vicinity of the amino acid substrate, all within the urzyme architecture, would not change specificities of subclass IB GlnRS to Glu [58,59] or subclass IC TrpRS to Tyr [60]. Changing GlnRS specificity to Glutamate [59] required wholesale mutations in the second layer surrounding the amino acid binding site outside the urzyme, but within in the GlnRS CP1 domain.

Similarly, a modular thermodynamic cycle comparing specificities of full-length TrpRS, urzyme, and urzyme plus either CP1 or the anticodon binding domain (ABD) showed rejection of Tyrosine by *G. stearothermophilus* TrpRS requires cooperation between CP1 and the ABD [33]. CP1 must coordinate its movements with those of the ABD to perform the allosteric communication necessary to enhance side chain selectivity beyond the modest capabilities of the urzyme [24,35]. In both cases, fine tuning specific recognition of amino acid substrates apparently required insertion of CP1 and the ABD and, subsequently, coupling between them. 

Inserting the CP1 motif would necessarily have disrupted bidirectional coding (Figure 1B), thus dividing aaRS evolutionary histories decisively into distinct stages (Figure 8, between orange and yellow backgrounds). Selective advantages of CP1 insertion thus appear to have been to (i) end constraints imposed by bidirectional coding and (ii) transcend the fundamental limitation on specificity posed by the urzyme architecture. Either or both would have allowed substantial, discontinuous, increases in fidelity to develop. CP1 therefore likely dramatically transformed the Class I aaRS fitness landscape, and was likely necessary to expand the coding alphabet.

### 3.6. A Revised Branching Order Suggests That Class I aaRS Protozymes and Urzymes Root the Rossmannoid Superfamily

Putting Class I aaRS protozymes at the root of that superfamily reconfigures many branching orders within the proteome to reflect that aaRS urzymes were not a late-developing branch in the Rossmannoid superfamily radiation, but instead were ancestral to it (Figure 9). Descent of Class I and II aaRS from a single bidirectional ancestral gene [14,22,23,25,29,40,41] would underscore the likelihood that the aaRS of both families diverged, rather than converging to similar functions from different sources. Thus, the genetic coding table itself likely evolved by bifurcating pre-existing aaRS genes into specialized enzymes whose more discriminating specificities for tRNA and amino acid substrates enabled daughter synthetases to differentiate groups amino acids that previously had functioned as synonymous [12,50,54,61].

It would be surprising if other branches of the proteome had not diverged from the ancestral aaRS, as suggested in Figure 9. 

## 4. Materials and Methods

Amino acid chemistry underlies the metric form of amino acid substitution matrices (aaSM) generally required for the logically circular process of creating an amino acid sequence alignment by optimizing the constraints provided by some aaSM and then building a phylogenetic tree from the resulting alignment according to those same constraints [12].

We superposed the available bacterial Class I aaRS crystal structures to base the multiple sequence alignment (MSA) exclusively on precise three-dimensional structural homology, freeing the MSA from dependence on empirical substitution matrices and estimates of relative rates of mutation [63]. In the first place, this was done within each aaRS type. The identification of sequence positions displaying very high conservation of both structure and amino acid occupancy provided unambiguous anchors for the production of much more extensive sequence alignments. Popinga [64] performed this task through extensive use of HHpred [65] and MODELLER [66] to thread amino acid sequences for each aaRS type using experimentally determined 3D structures as templates. This procedure produced, for each aaRS type, an expanded MSA in which various well-conserved secondary structural regions were near-perfectly structure- and sequence-aligned between species known to have diverged not long after the LUCA. 

In the final stage of alignment, the structures of the individual aaRS types were brought together to identify regions of universal structural homology, along lines similar to those described by Chaliotis [67], to produce what we refer to as the Class I aaRS “scaffold”. The product was an MSA across all Class I aaRS types in which each sequence position could be assumed to have arisen, as far as is reasonably possible, from the same LUCA-ancestral codon. While the validity of this assumption is by its very nature untestable, the close structural homology of Class I aaRS of all types and the rigor of our procedure gives us confidence and justification as good as that underlying any cross-species phylogenetic enterprise. However, we took further steps to restrict and constrain the data upon which we later built putative phylogenies. First, individual scaffold elements are more extensive in some extant aaRS types than others and it is not possible to identify proper residue-by-residue homology among the loops, turns and structurally disordered regions that join them. All these were excluded from the Class I “scaffold” MSA. Second, the earliest domain of life is unknown and highly controversial, without consensus [68,69,70]. However, we included only eubacterial aaRS for the following reasons.

The substantially stronger codon middle-base pairing frequency and the steeper slope between independently reconstructed ancestral eubacterial Class I and II sequences ([40]; Chandrasekaran, unpublished data; Carter & Wills unpublished data) provide evidence that bacterial aaRS sequences are both closer to the ancestral root and less convoluted by horizontal gene transfer than those from other domains. Analysis of loop structures is a contested issue; however there is certainly no consensus that they contain reliable information about the evolutionary origin of biology’s three main evolutionary domains [71,72,73,74,75]. 

It is generally accepted that cytoplasmic proteins of bacteria have been subject to many fewer complex selection pressures than their archeal and eukaryotic counterparts. The complex functional and regulatory roles played by some aaRS proteins and their involvement in numerous genetic syndromes attests to this [76].

Thus, the final MSA contained roughly bacterial 20 sequences for each of 10 Class I aaRS, circumventing as many problems as possible in aligning sequences that diverged to produce different substrate specificities in the pre-LUCA æon. Use of scaffold sequences for phylogenetic analysis gave the best guarantee that results would reflect relatively neutral evolution within the context of asymmetric, specialized selection pressures producing variant substrate specificities by fine-tuning the size, shape and chemistry of more intricately constructed amino acid side-chain structures and pockets.

The MSA for the Class I scaffold was output using VMD [77], and is provided in the Appendix A together with all subsets used in this work. The scaffold fasta file was then partitioned along lines of the experimental deconstruction of the Class I aaRS superfamily [25] into the disjointed segments of the urzyme, separated by structurally conserved segments from CP1. Finally, because the urzyme (83) and CP1 (20) partitions of the scaffold MSAs have different sequence lengths, seven subsets comprising 20 sequence positions distributed throughout the urzyme were selected arbitrarily by a balanced, randomized procedure [78] to test the effect of sequence length on our phylogenetic signatures. We invoked the usual “zeroth order”assumption that the evolution of any sequence position was statistically independent of all other positions.

Phylogenetic analysis involves construction and analysis of family trees that represent the histories of related evolving objects. Our trees were constructed using the Markov Chain Monte Carlo procedures implemented in the widely available BEAST2 program [79] (https://github.com/CompEvol/beast2/ accessed on 2 February 2019). Statistical data accumulated by most commonly used phylogenetics platforms during the Bayesian search for the most probable trees furnish complementary insight about why the resulting ensemble represents the most probable trees. These statistics include the tree height, a shape parameter, and the clade support. A tutorial is available introducing the use of the BEAST2 program and its applications [80].

The Tree Height reflects the inverse of the mean overall mutation rates. Site-specific mutation rates are fitted to a gamma distribution [81] by adjusting the Shape parameter, α = 1/CV^2^, where CV is the coefficient of variation or the ratio of the standard deviation to the mean. Shape is thus a measure of Tree Height variance. Shape is commonly regarded as a ‘nuisance’ parameter. It is adjusted to ensure proper mixing and convergence of the MCMC search. It is seldom, if ever, of further interest. The Tree Height parameter, however, is very commonly used in phylolinguistics to estimate rates at which individual features change during the growth of tree branches (see, for example, [82,83]). 

Clade support, *S_i_*, is the percentage of all trees for which the aaRS type in question appears to be monophyletic, meaning that the leaves of that aaRS type descend from the same most recent branch point (common ancestor) with no descendants arising from a different aaRS type. Because clade support can be reduced by the presence of common features that do not arise from common ancestry [84], this metric is useful in distinguishing between different hypotheses about the structure of a tree. It is, for example used to distinguish between various hypotheses about the reconstruction of Trans-Eurasian languages [83].

The conservation quality, *Q_j_*, defined by Clustal [43,44] was computed down each column, *j*, of the grand MSA, which included all aaRS types. While the definition of *Q* seems convoluted, it has been constructed as a metric whose value reflects the degree of amino acid diversity generated by typical evolutionary amino acid substitution processes, reflected in the substitution matrix that it uses as a reference. Different matrices do not give widely divergent *Q* parameters. We used the ClustalW default matrix (PAM 250; [85]) and calculated the average, $\langle Q\rangle$, over all positions within a partition of the MSA a parametric representation of the partition-wide variation in amino acid occupancy calculated column-by-column over individual sequence positions. In the course of the analysis it emerged that $\langle Q\rangle$ is actually nearly co-linear with the log(Shape) parameter (Figure 4).

A third, row-by-row metric was derived from molecular kinships between rows of the MSA within each partition by clustering the sequences into clades according to their evolutionary origin. Phylogenetic trees were computed using BEAST2 [79], allowing the use of multiple amino acid substitution matrices (primarily WAG and LG). Trees were visualized with DensiTree ([86]; https://github.com/rbouckaert/DensiTree accessed 5 July 2019) and FigTree ([87]; https://github.com/rambaut/figtree accessed 2 February 2019). For the full Class I scaffold and each partition of interest, DensiTree was used to extract ~10,000 trees generated by BEAST2 in building trees. From these we calculated values of a parameter representing the support for clades that each corresponded exclusively to an individual aaRS type, i. 

The mean clade support, $\langle S\rangle$, is a metric derived from a row-by-row calculation over all 10,000 phylogenetic trees generated using the sequence data for a chosen partition of the scaffold MSA. Cross correlation between the three metrics is negligible (Table 2), so they reflect independent aspects of the MSAs.

We compiled three complementary metrics to compare the overall scaffold MSA to its Class I urzyme, protozyme, and CP1 subsets. 

Columns for various predictors were appended to the table of $\langle Q\rangle \;$and $\langle S\rangle$ values to form the design matrix (Table 2) used for multiple regression analysis with the JMP software [88]. Multiple Regression was used to assess the statistical strength of contributions of the predictors (i.e., the independent variables, A, B, C, CP1 defined in Figure 1 and the number of amino acids) to $\langle Q\rangle \;$and $\langle S\rangle$ values, using stepwise searches to identify the best set of predictors followed by least squares estimation of the corresponding coefficients and their Student *t*-test *p* values.

## 5. Conclusions

Understanding how evolution of aaRS·tRNA cognate pairs effected the stepwise increases in dimensionality of entries into the universal genetic coding table is pivotal to the origin of biology. To that end we have sought relevant data from carefully curated, structure-based amino acid sequence alignments (MSAs) of secondary structures shared by all members of the Class I aaRS superfamily. Three uncorrelated phylogenetic metrics identify significant, high-resolution mosaicity, consistent with assembly from distinct genetic sources. All metrics reinforce prior arguments that Class I aaRS evolved by a succession of intermediate states—protozyme=>urzyme=>Catalytic domain—with increasingly sophisticated catalytic [14,23,25,29,39] and discriminatory [50,54,89] capabilities. Regression analyses identify covariation of sequences as statistically significant two-way intermodular interactions that facilitate functional interpretations and help validate our approach. 

Genetic coding and the proteome probably emerged from an RNA·polypeptide partnership. Previously accepted phylogenies of the Class I aaRS root well before the Last Universal Common Ancestor (LUCA) [7,11,18,19,47], but appeared also to have radiated after earlier bifurcations in the immense meta-family [7] containing folds based on the Rossmann dinucleotide-binding fold [90]. Genetic coding by protein aaRS was therefore thought, necessarily to have replaced a prior implementation based on ribozymal assignment catalysts [62,91]. This paradoxically late radiation of protein assignment catalysts has been the foremost phylogenetic evidence favoring the RNA World hypothesis. 

Evidence described here that CP1 is a more recent acquisition by ancestral aaRS urzymes supports a plausible alternative branching order (Figure 8). Attributing the apparently late adaptive radiation of Class I aaRS to CP1 achieves consistency with an origin of genetic coding from a bidirectional gene administering a binary coding alphabet in a peptide/RNA world. Resolving the paradox in this way complements—without necessarily contradicting—the phylogenetic analyses of Koonin [62,91]. 

The Bayesian phylogenetic evidence adduced here for high-resolution mosaicity within protein domains has no precedent. Our observations suggest that this work points toward more substantial applications of software like BEAST2 [79] to a broader range of evolutionary questions in other protein superfamilies.

## Figures and Tables

**Figure 1 ijms-23-01520-f001:**
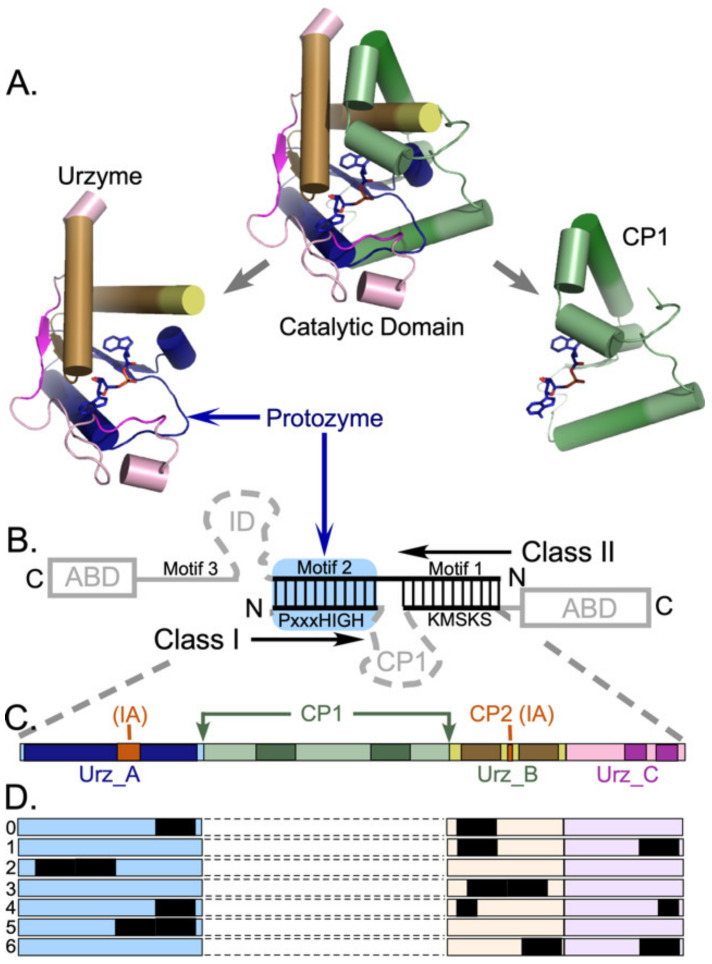
Structural modules underlying the hypothesis and data organization. (**A**) Deconstruction of Class I urzyme and internal connecting peptide 1 (CP1) [20] insertions that together make up the Class I catalytic domain. Cartoons were prepared with Pymol [21] from coordinates (PDB ID 1I6L) for the tryptophanyl-tRNA synthetase, the smallest Class I aaRS. Secondary structures displayed here are conserved in all 10 Class I aaRS. The activated aminoacyl adenylate is drawn as sticks. Size variation in other Class I aaRS arises from further insertions within CP1. Anticodon-binding domains are idiosyncratic, and not shown. (**B**) Overlapping portions of Class I and II aaRS as envisioned in an ancestral bidirectional gene [22] coincide with the respective urzymes [17,23]. Vertical lines denote ancestral base-pairing between the respective genes. Grey segments were presumably more recent additions. Note that, because insertion of CP1 is incompatible with bidirectional coding, the structural conservation of secondary structures within CP1 (dark green) is unexpected. (**C**) Schematic location of CP1 between two roughly equal fragments of the urzyme, colored to allow identification of structural fragments in (**A**) Intensely colored segments are highly conserved secondary structures in all 10 Class I aaRS and compose the Class I scaffold. Subclass I A enzymes contain one or more additional insertions (red). (**D**) Sampled alignments consisting of different segments totaling 20 amino acids, selected as described in Methods. (**C**,**D**) elaborate the Class I base-paired portion of (**B**) The protozyme (Urz_A) is the amino terminal β-α-β crossover connection (blue), the amino acid binding pocket (Urz_B) is formed by the protozyme and two intermediate α-helices (amber). The KMSKS loop (Urz_C), which binds the pyrophosphate leaving group, is rose. Previous work [24] established that the CP1 insertion (green) has little impact on the enzymatic properties of the TrpRS urzyme unless the anticodon-binding domain is also present. Colors match those in (**A**).

**Figure 2 ijms-23-01520-f002:**
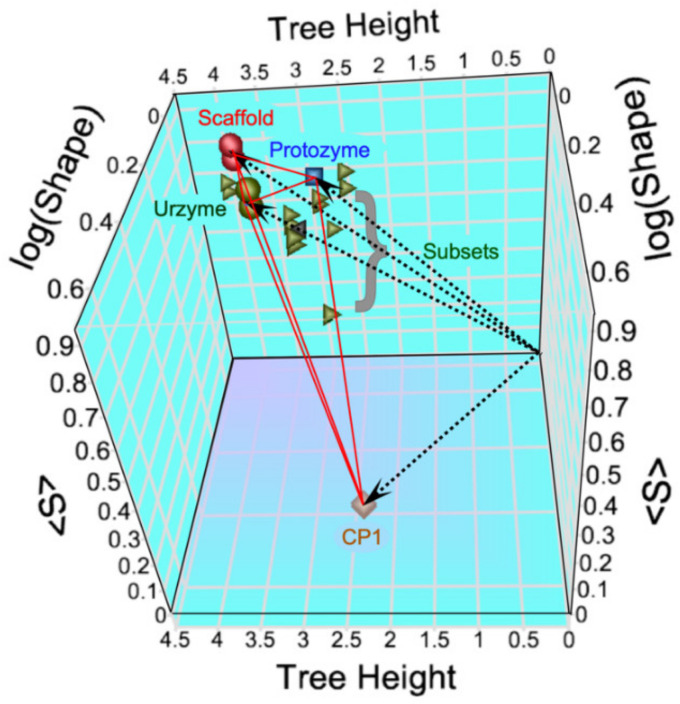
The different modular MSAs for the entire (eubacterial) Class I aaRS superfamily are widely separated in the vector space spanned by the three linearly independent phylogenetic metrics. The dashed lines connect the centroids of each MSA to the origin in the rear right-hand bottom corner. Red spheres are the entire scaffold, green spheres are the urzyme, blue squares are the Protozyme, and brown diamonds are the CP1 sequences. Green triangles are seven different random subsets of the Urzyme. The base plane (Tree Height and log(Shape)) spans the column-by-column metric and its variance. The vertical axis spans the row-by-row clade support, $\langle S\rangle$. Both WAG and LG values are shown, accounting for the doubling of symbols.

**Figure 3 ijms-23-01520-f003:**
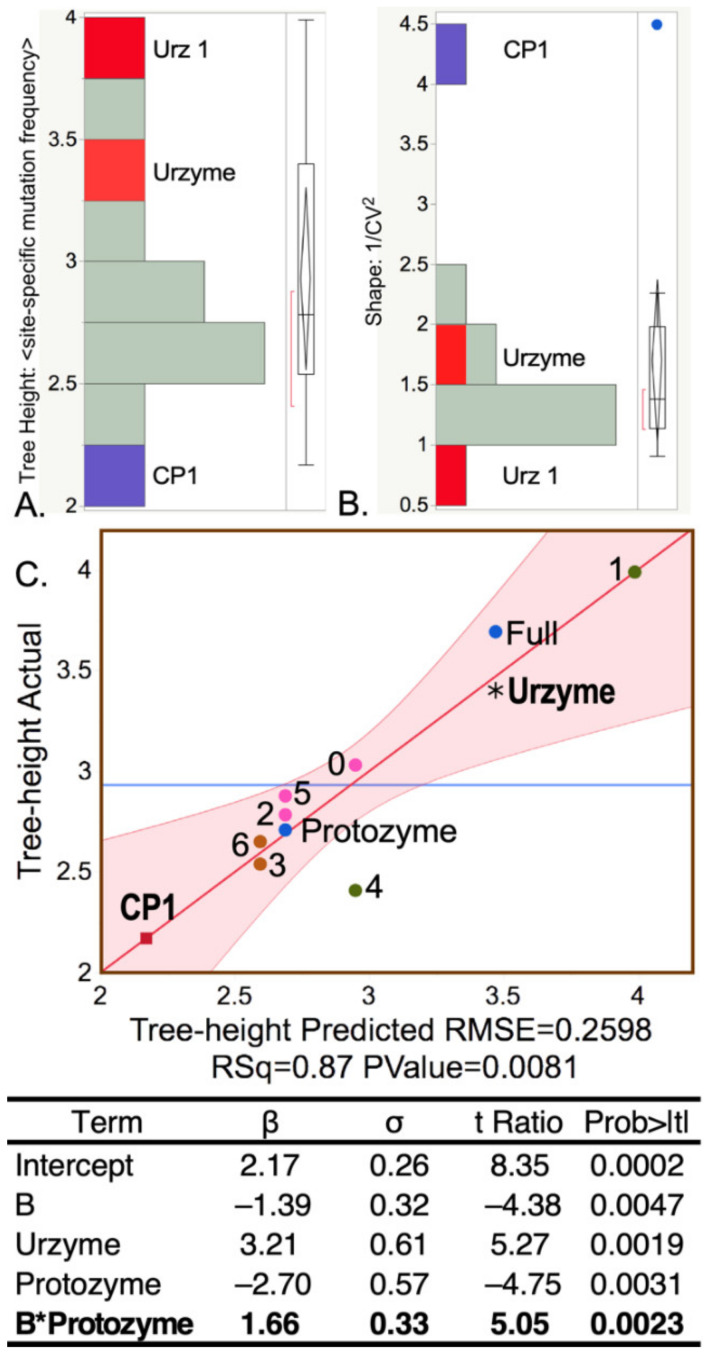
Histograms of the Tree Height (**A**) and Shape (**B**) metrics highlight differences between sequence variability within CP1 (blue) and urzyme (red) sequences. The Tree Height is the reciprocal of the estimated mutation rate per site. CP1 has the lowest site-specific mutation rate and the highest Shape parameter (blue dot implies statistical significance). (**C**) Regression model showing the dependence of Tree Height on MSA partitions, together with a table of regression coefficients (β) and their standard deviations (σ), Student *t*-tests, and their probabilities. The Urz_1 subset of amino acids includes a stretch of 10 amino acids along the specificity determining helix, where the highest site-specific mutation frequency occurs within the urzyme, putting it in the upper right corner. The horizontal blue line in this and other regression curves is the average value of the dependent variable. Datapoints with different shapes represent the various MSAs in the design matrix and are colored and labeled for identification. Numerals refer to the 20 residue subsets defined in Figure 1. R^2^ for this model is 0.87. The regression table provides the β coefficients of the best regression model, together with their standard deviations (σ) and *t*-test values and their *p*-values under the null hypothesis. The F-ratio is 9.9, with a *p*-value of 0.008. Here and elsewhere, two-way interactions are in boldface. Their functional significance is discussed in Section 2.4.

**Figure 4 ijms-23-01520-f004:**
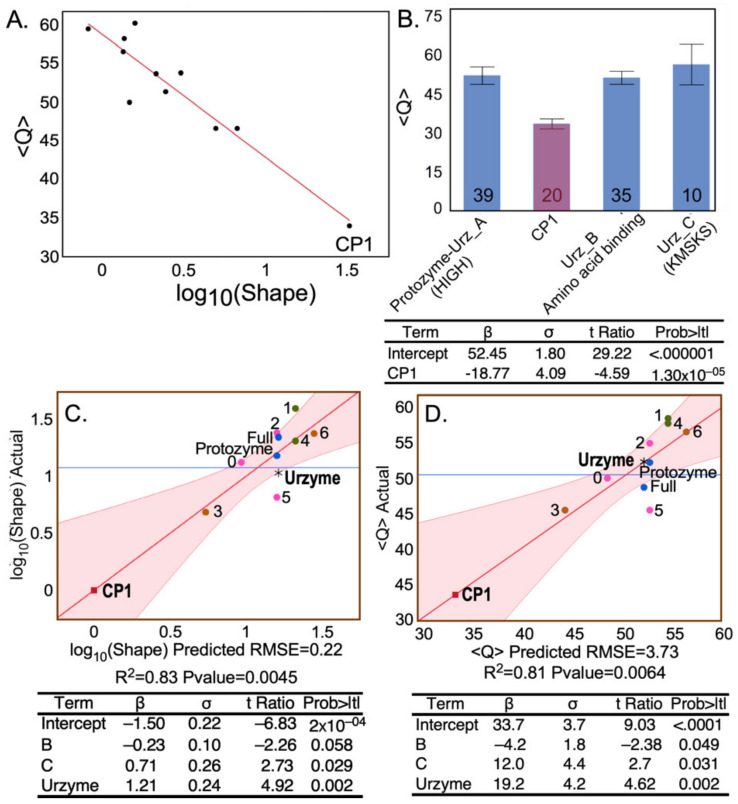
Position-specific metrics. (**A**) Colinearity of the logarithm of the Shape parameter for the gamma distribution of the Tree Heights estimated by BEAST2 for different MSAs derived from the Class I scaffold, and the mean conservation quality scores, $\langle Q\rangle$ [44], obtained by Clustal directly from respective MSAs. (**B**) Comparison of $\langle Q\rangle$ for the four segments of the Class I scaffold alignment. Class I signature peptides [4] are in parentheses. Error bars show the standard error of the mean overall positions within the segment. The numbers of amino acids in each segment are given for each histogram. (**C**,**D**) Regression models in (**C**,**D**) have β coefficients and σ values showing nearly identical dependence of –log(Shape) and $\langle Q\rangle$, respectively, on the same predictors from the design matrix (Table 1). Dots represent different MSAs and are labeled and colored as in Figure 3 to emphasize the extraordinary similarity of column-by-column metrics derived in two different ways.

**Figure 5 ijms-23-01520-f005:**
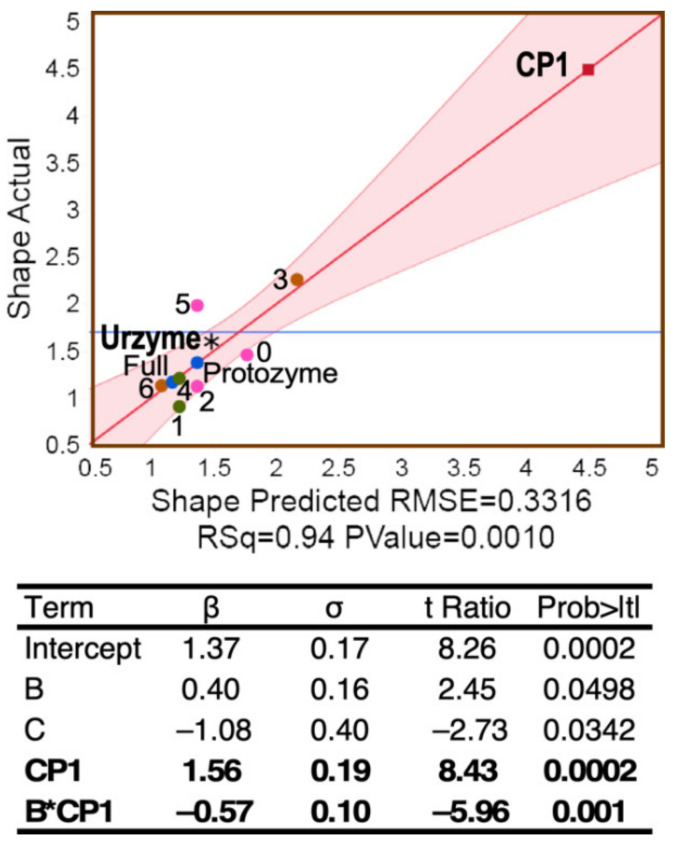
Shape parameter dependence. Plot of the multivariate regression model for Shape. R^2^ = 0.94 for the regression and the *p* value for the F ratio is 0.0006. The table provides regression coefficients, β, their standard deviations, σ, and their student *t*-test probabilities. Significant predictors of Shape are the amino acid positions in CP1, those in the B-fragment containing the amino acid specificity-determining helix (sand; Figure 1A), and their two-way interaction, as well as the C-fragment.

**Figure 6 ijms-23-01520-f006:**
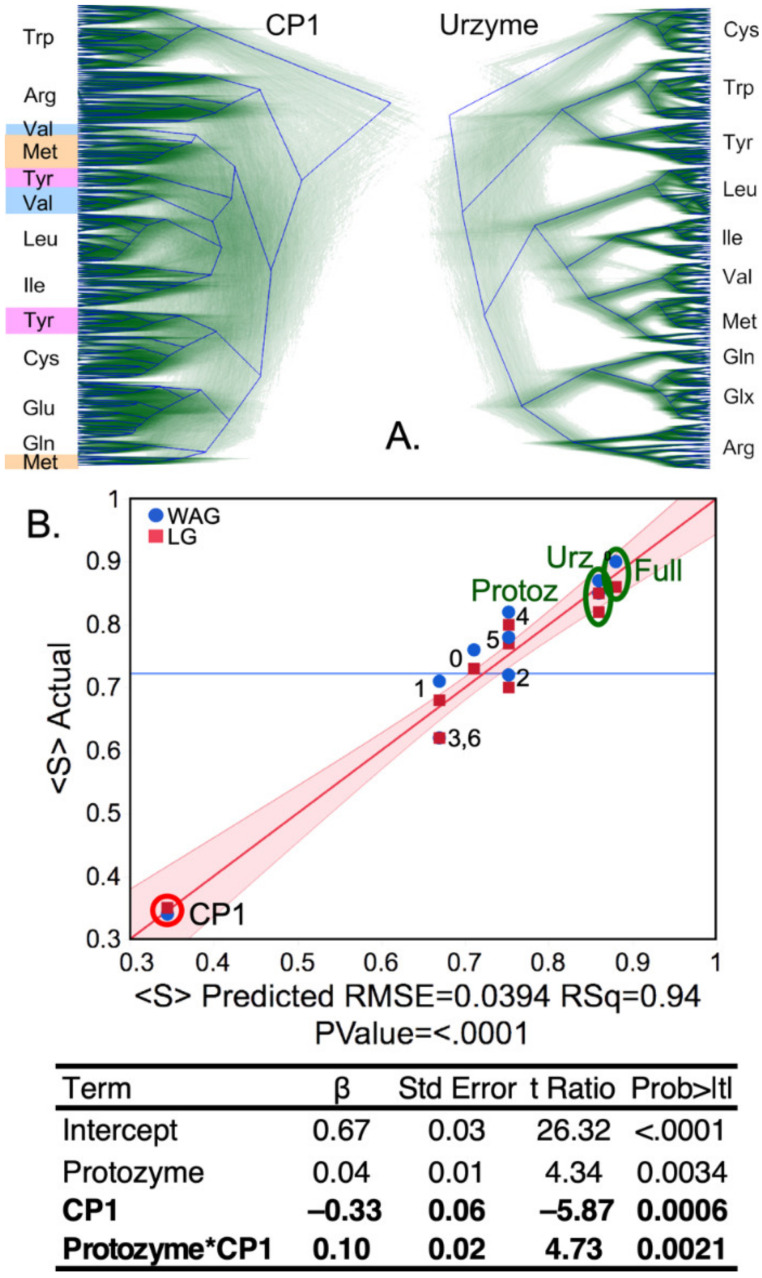
Phylogenetic support. (**A**). DensiTree representations of the urzyme and CP1 segments of Class I structural scaffold. Each aaRS is monophyletic in the urzyme alignment, whereas several of the clades in the CP1 alignment, highlighted in color, have multiple ancestries. (**B**). Regression model for $\langle S_{i}\rangle$. Dots are colored and labeled as in Figure 3. β coefficients and σ values in the regression table are for the WAG matrix, as the use of both WAG and LG matrices unduly enhances the *p*-values of the β-coefficients. The R^2^ and *p* value of the model’s F-ratio are shown under the *X*-axis. Coefficients and their statistics for the model are given in Table 2. Individual $\langle S\rangle$ values are labeled to enable comparison with Figure 1C. R^2^ was 0.94 for 20 observations, and the F-ratio for the regression table was 92.4 with a *p*-value < 0.0001.

**Figure 7 ijms-23-01520-f007:**
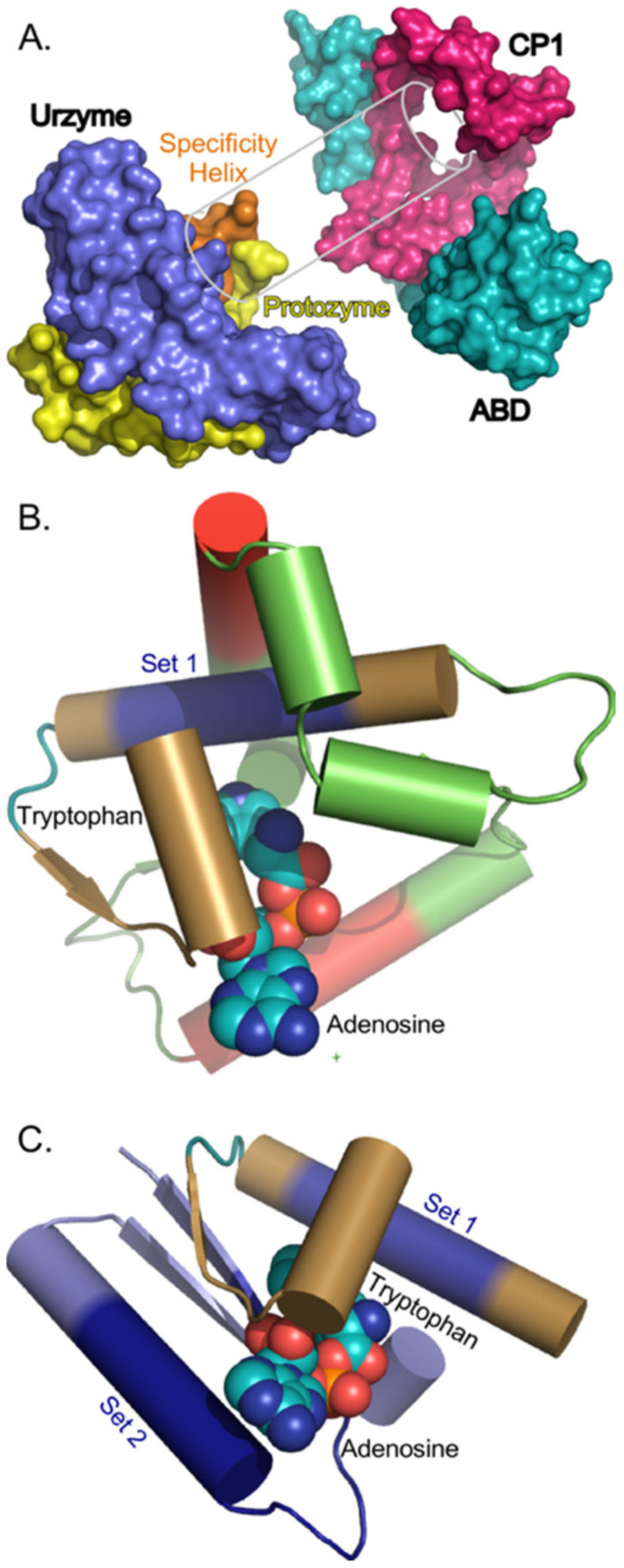
(**A**) Structural relationships between the TrpRS urzyme, CP1, and the anticodon-binding domain (ABD). Coloring differs from that in Figure 1. The CP1 motif forms an annulus that constrains motions of the specificity determining helix (sand) and the Protozyme (yellow), constraining, in turn, the effective size of the amino acid-binding pocket when the ABD (teal) changes its orientation. Experimental evidence [33] described in the text confirms that these constraints enable full-length TrpRS to reject tyrosine in the transition state complex for tryptophan activation. (**B**) Structural cartoon with details of the interaction illustrated in (**A**). Coloring is the same as that in Figure 1. The specificity-determining helix is the site of the Set 1 segment, which is colored in blue. The TrpRS CP1 module is green, and the scaffold segments are red. Note the close contact between CP1—especially the red scaffold segment—and Set 1. (**C**) Interactions between the TrpRS protozyme, locus of the ATP binding site, and segment B, locus of the amino acid binding site. Subsets 2 and 1, which are contained entirely within the respective segments, are highlighted in dark blue.

**Figure 9 ijms-23-01520-f009:**
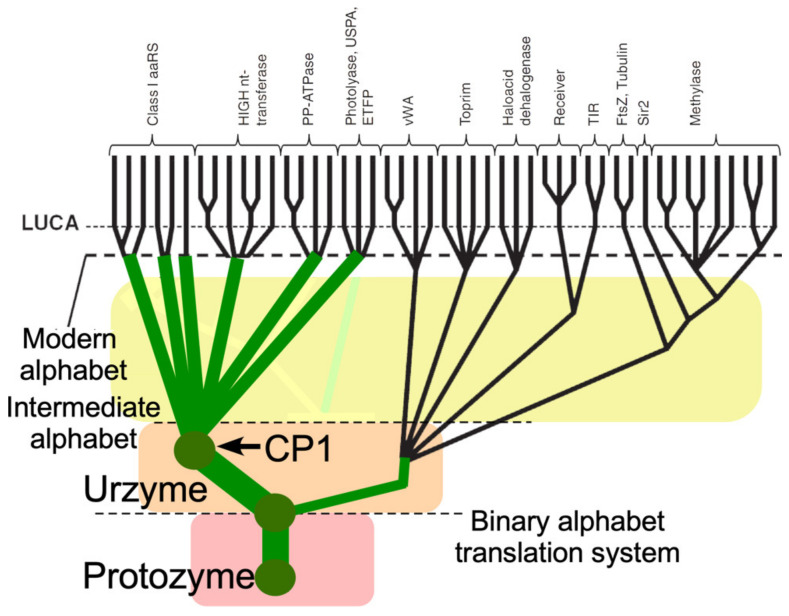
Modified Class I aaRS phylogeny consistent with bidirectional coding ancestry in a peptide/RNA world. Adapted with permission from Figure 12.2A of [62]. Green circles represent key stages in the emergence of genetic coding, beginning with ATP-dependent amino acid activation by protozymes coded by a bidirectional gene [14]. Backgrounds are colored as in Figure 8B to approximate the expansion of the proteome possible with suggestive increases in the dimension of the alphabet.

**Table 1 ijms-23-01520-t001:** Design matrix for regression analyses. Shaded columns are dependent variables; unshaded columns are independent variables to be tested as predictors by building regression models. ${\langle S\rangle }_{WAG}$ and ${\langle S\rangle }_{LG}$ are the mean clade support in the ensemble of trees built according to the WAG and LG amino acid substitution matrices. As discussed in the text, they are nearly identical. Numerals in columns A, B, C, and CP1 are proportional to the total number of amino acids in the respective MSAs. NUMB is the number of amino acids in the alignment. Two additional independent variables, urzyme and protozyme, constructed in a related fashion, are not shown.

Subset	<*S*>_*WAG*_	<*S*>_*LG*_	<Q>	Tree Height	Shape	A	B	C	CP1	NUMB
Full	0.9	0.86	48.8	3.69	1.17	4	3	1	2	103
urzyme	0.87	0.82	52.4	3.40	1.60	4	3	1	0	83
Protoz	0.85	0.85	52.3	2.71	1.38	4	0	0	0	46
CP1	0.34	0.35	33.7	2.17	4.49	0	0	0	2	20
Urz_0	0.76	0.73	50.1	3.03	1.46	1	1	0	0	20
Urz_1	0.71	0.68	57.8	3.99	0.91	0	1	0.5	0	20
Urz_2	0.72	0.7	55	2.78	1.13	2	0	0	0	20
Urz_3	0.62	0.62	45.6	2.54	2.26	0	2	0	0	20
Urz_4	0.82	0.77	58.5	2.41	1.21	2	1	0.5	0	20
Urz_5	0.78	0.8	45.6	2.88	1.98	4	0	0	0	20
Urz_6	0.62		56.6	2.65	1.13	0	2	1	0	20

**Table 2 ijms-23-01520-t002:** Cross-correlation R^2^ values between phylogenetic metrics (excluding CP1).

	Tree Height	Shape
Shape	0.15	
$\langle S\rangle$	0.10	0.03

## Data Availability

All data used in this work are available in the Appendix A. Links are provided in the text or software used in the study.

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
