# Peer review of "Multidimensional Phylogenetic Metrics Identify Class I Aminoacyl-tRNA Synthetase Evolutionary Mosaicity and Inter-Modular Coupling"

_ijms, 2022, doi:10.3390/ijms23031520_

Round 1
Reviewer 1 Report
Aminoacyl-tRNA synthetases (aaRS) are very attractive targets for potential drug development as well as biological evolution.
The authors introduced Evolutionary Mosaicity and inter-modular couping for aminoacyl-tRNA synthetases (aaRS) through computational work.
The authors develop multi-dimensional phylogenetic metrics from the ensemble of phylogenetic trees obtained by Markov Chain Monte Carlo (MCMC) simulations.
In my personal opinion, the sources for these computational approaches, if programmed, should be open and verified to be trusted by all researchers. If the general researcher does not have access to the author's development, the general researcher cannot verify that the author's evolutionary classification is correct. I am very interested in the research introduction, but I have doubts about how far to trust these results.
1. The source of the program developed by the author should be published in Zenodo, Github or ...
2. ORCID is written after the author's address, which is different from the journal format. Also, there were 4 authors, but only 3 ORCIDs were written.
3. Supplement_MSA_files.txt does not exist in the MDPI web.
4. Displaying figures in subtitles is not appropriate.
2.4.1. CP1 forms a structural annulus constraining the urzyme’s two halves (Figure 6A).
2.4.2. The Tree Height dependence on segment B (Figure 2C) changes sign, depending on whether 410 the protozyme is present.
2.4.3. The Shape dependence on segment B (Figure 4) changes sign, depending on whether CP1 is 427 present.
Line 412: "Figures 2A&B" should be "Figures 2A and 2B"
Line 422: "Figure 6B&C" should be "Fgiures 6B and 6C)
In Figure 2, it is consistent that "A and B" appear at the top of the figure.
Author Response
Both reviewers identified problems that clouded our intended meaning in this manuscript, motivating a thorough revision along lines they suggest. The resulting manuscript is therefore stronger, more linearly argued, and less ambiguous on important details. We are grateful for their remarks. Detailed responses to specific concerns are interspersed below in 12 point Georgia font.
Reviewer #1
The authors develop multi-dimensional phylogenetic metrics from the ensemble of phylogenetic trees obtained by Markov Chain Monte Carlo (MCMC) simulations.
In my personal opinion, the sources for these computational approaches, if programmed, should be open and verified to be trusted by all researchers. If the general researcher does not have access to the author's development, the general researcher cannot verify that the author's evolutionary classification is correct. I am very interested in the research introduction, but I have doubts about how far to trust these results.
We are grateful for the expression of high interest in the work, and concur with the reviewer with regard to facilitating reproduction of our results.
In addition, we have extensively revised discussion in Section 2.4 to enhance the clarity of correlations the phylogenetic metrics and experimental structural and functional data, to underscore the justification of what we report.
- The source of the program developed by the author should be published in Zenodo, Github or ...
Perhaps the reviewer infers from our presentation that we developed novel software platforms to carry out the work described in the manuscript. As we note in the new introductory section of Results, our work only makes novel use of metrics already provided in standard phylogenetics software. We have changed misleading language in the introduction (lines 210-271) that could have led to such misreading.
In fact, the novelty in our ms is only the analytical use to which we put the quantitative metrics accumulated during phylogenetic tree construction, which are provided by virtually all phylogenetics programs that estimate dating. All software used in this ms is publicly available, as noted in the literature citations. We have included the github url for BEAST2 (page 22; line 785) in new paragraph of the Methods section in which we clarify our procedures and give a reference to a useful tutorial on the use of BEAST2. Urls for densitree and figtree are given on lines 826 and 827 of the revision.
Reviewer 1’s concern here underscores the fact that our original submission may not have been entirely clear about what we have done. To address that possibility, we also have added new paragraphs at the beginning of the results on page 4, lines 173-213, including a new figure (Figure 2), to clarify what we have done with some precision.
ORCID is written after the author's address, which is different from the journal format. Also, there were 4 authors, but only 3 ORCIDs were written.
We have sought the advice of MDPI on this point and been advised that we can retain the ORCID IDs for authors who have them.
Supplement_MSA_files.txt does not exist in the MDPI web.
The MSA files used for the phylogenetic analysis are all provided in a supplement submitted at the time of the original submission. It is not our responsibility to assure that this supplement is not made available to reviewers. The supplemental file will be available to readers, and we believe that the editorial staff could have provided the supplemental file upon request from the reviewers.
Displaying figures in subtitles is not appropriate.
We have removed citations of figures in the following subheadings:
2.4.1. CP1 forms a structural annulus constraining the urzyme’s two halves (Figure 6A).
2.4.2. The Tree Height dependence on segment B (Figure 2C) changes sign, depending on whether 410 the protozyme is present.
2.4.3. The Shape dependence on segment B (Figure 4) changes sign, depending on whether CP1 is 427 present.
Line 412: "Figures 2A&B" should be "Figures 2A and 2B"
Line 422: "Figure 6B&C" should be "Fgiures 6B and 6C)
These changes have been made in the revision
In Figure 2, it is consistent that "A and B" appear at the top of the figure.
We have revised Figure 2 (now Figure 3) accordingly.
Reviewer 2 Report
Determining the evolutionary history of t-RNA synthetases is key to understanding how the genetic code for protein synthesis came into being and thus is crucial in understanding abiogenesis. The use of structural and sequence alignments in conjunction The authors identify a scaffold shared by Class I t-RNA synthetases and provide evidence that the scaffold is a mosaic assembled from multiple genetic sources. They further postulate that a minimal catalytic fragment of the enzyme (protozyme) forms the root of the class and subsequently acquired additional functional genetic elements. The authors suggest that these findings are phylogenetic support for an evolutionary trajectory wherein primitive enzymes/catalysts are successively improved upon by adding small functional elements, forming the basis for the emergence and refinement of the genetic code. The manuscript is of good quality and is suitable for publication in IJMS. However there are a few typographical errors and hence careful proofreading is suggested prior to publication. Some instances of errors and additional suggestions are listed below, this does not cover all instances of similar errors.
Abstract -
Line 33 - (Typographical error, remove duplicate punctuation in the end of sentence) establishes a timeline for the growth of coding from a binary amino acid alphabet.
Introduction - The concepts of Urzyme and protozyme are still rather esoteric and need to be introduced adequately for a general reader. The explanations provided in text (Line 88 and lines 95-97) are still vague and raises questions with respect to the actual definition of these terms. The literature referred (also co-authored by Carter, C.) provides a clearer definition, but they need to be mentioned here.
Line 67 - The abbreviation “CP1” is used without defining the expansion previously. The expansion is provided only subsequently in line 114.
Line 125 (Typographical error) - Evidence for descent of Class I and II aaRS from a bidirectional gene implies.
Table 1 - Include definitions for variables <S>LG and <S>WAG
Line 220 - Format <Q> to match the rest of the text
Line 681 (Typographical error) - along lines similar to those described by Chaliotis [69], to produce what we refer
Author Response
Reviewer #2
Both reviewers identified problems that clouded our intended meaning in this manuscript, motivating a thorough revision along lines they suggest. The resulting manuscript is therefore stronger, more linearly argued, and less ambiguous on important details. We are grateful for their remarks. Detailed responses to specific concerns are interspersed below in 12 point Georgia font.
Determining the evolutionary history of t-RNA synthetases is key to understanding how the genetic code for protein synthesis came into being and thus is crucial in understanding abiogenesis. The use of structural and sequence alignments in conjunction The authors identify a scaffold shared by Class I t-RNA synthetases and provide evidence that the scaffold is a mosaic assembled from multiple genetic sources. They further postulate that a minimal catalytic fragment of the enzyme (protozyme) forms the root of the class and subsequently acquired additional functional genetic elements. The authors suggest that these findings are phylogenetic support for an evolutionary trajectory wherein primitive enzymes/catalysts are successively improved upon by adding small functional elements, forming the basis for the emergence and refinement of the genetic code. The manuscript is of good quality and is suitable for publication in IJMS. However there are a few typographical errors and hence careful proofreading is suggested prior to publication. Some instances of errors and additional suggestions are listed below, this does not cover all instances of similar errors.
We thank the reviewer for the encouraging comments and for the clear statement of our goals and results. We have revised the ms extensively, re-doing many of the figures to enhance clarity. All revisions are highlighted in the separate file with tracked changes.
Abstract -
Line 33 - (Typographical error, remove duplicate punctuation in the end of sentence) establishes a timeline for the growth of coding from a binary amino acid alphabet.
We correct this infelicity in the revision.
Introduction - The concepts of Urzyme and protozyme are still rather esoteric and need to be introduced adequately for a general reader. The explanations provided in text (Line 88 and lines 95-97) are still vague and raises questions with respect to the actual definition of these terms. The literature referred (also co-authored by Carter, C.) provides a clearer definition, but they need to be mentioned here.
We concur with the reviewer, and have amplified the description of urzymes in lines 90-103 on page 3. Introduction of the protozyme is supplemented in lines 111-114, hinting at the more important conclusions drawn from the rest of the Results, and in lines 115-118. We also revised Figure 1 to include designation of the protozyme.
Line 67 - The abbreviation “CP1” is used without defining the expansion previously. The expansion is provided only subsequently in line 114.
Reference to the full name and its original literature citation have been added to line 74, and the introduction in the text has been moved closer to Figure 1 in the third paragraph of page 3.
Line 125 (Typographical error) - Evidence for descent of Class I and II aaRS from a bidirectional gene implies.
We thank the reviewer for catching this error, which is corrected in the revision.
Table 1 - Include definitions for variables <S>LG and <S>WAG
The two mean support variables are defined in the Table 1 caption (lines 216-217 as well as in the text on lines 177-178 in the first paragraph of Results.
Line 220 - Format <Q> to match the rest of the text
We have tried to make the formatting of the dependent variables <S> and <Q> consistent with their appearance in the equation settings thoughout the revised ms, except for the entries in the header line of Table 1, which we have left as is to avoid formatting problems.
Line 681 (Typographical error) - along lines similar to those described by Chaliotis [69], to produce what we refer
We thank the reviewer for calling attention to this and numerous other typographical errors, which we have tried to identify and correct in the revision.
Round 2
Reviewer 1 Report
The authors addressed the concerns of the reviewers. However, I have a question about whether the expression 'high-resolution' in the title is appropriate. I support the publication of this manuscript and suggest omitting 'high-resolution' for minor revisions.
Minor
Authors should revise the references according to the journal format.
Author Response
We have deleted "High-Resolution" from the title and meta-data on age 1 as requested.
References have been reformatted according to the IJMS styles panel as endnote footnotes.
We also have simplified the graphical abstract and revised the Abstract to ensure compatibility between it and the graphical abstract to make the latter more self-explanatory.